# Epidemiological and Clinical Characteristics of Respiratory Syncytial Virus Infections in Children Aged <5 Years in China, from 2014–2018

**DOI:** 10.3390/vaccines10050810

**Published:** 2022-05-20

**Authors:** Hangjie Zhang, Aiqing Zhu, George F. Gao, Zhongjie Li

**Affiliations:** 1NHC Key Laboratory of Biosafety, National Institute for Viral Disease Control and Prevention, Chinese Center for Disease Control and Prevention, Beijing 102206, China; hjzhang@cdc.zj.cn; 2The Center for Disease Control and Prevention of Zhejiang Province, Hangzhou 310051, China; 3Chinese Field Epidemiology Training Program, Chinese Center for Disease Control and Prevention, Beijing 102206, China; lizj@chinacdc.cn; 4Key Laboratory of Surveillance and Early-Warning on Infectious Disease, Division of Infectious Disease, Chinese Center for Disease Control and Prevention, Beijing 102206, China; zhuaq@chinacdc.cn

**Keywords:** respiratory syncytial virus, epidemiology, clinical, surveillance, child

## Abstract

Respiratory syncytial virus (RSV) is an important pathogen that causes acute respiratory tract infections in children. To understand the epidemiological and clinical characteristics of RSV in children, we analyzed the RSV diagnostic testing results from the 2014–2018 surveillance of acute respiratory infections in China. Among children aged <5 years, RSV incidence during 2014–2018 was 17.3% (3449/19,898), and 89.1% of RSV-positive individuals were inpatients. Children aged < 6 months had a high proportion in RSV infected individual (*n* = 1234; 35.8%). The highest RSV detection rate was in winter, RSV-A and RSV-B co-circulated year-round and jointly prevailed in 2015–2016. Cough was the common symptom of RSV infection 93.2% (3216/3449). Compared with older children, those aged <6 months were more likely to show breathing difficulty or lung rale that further developed into bronchopneumonia (*p* < 0.001). The symptoms such as cough, running nose, and diarrhea had significant differences between the RSV-A and RSV-B groups. The rate of RSV co-detection with other viruses or bacteria was 35.4%. Those coinfected with other viruses had a significantly higher incidence of fever, whereas those coinfected with bacteria had higher incidences of breathing difficulty and severe pneumonia. Our findings highlight the need for accumulating epidemiological information for the prevention and control of RSV.

## 1. Introduction

Respiratory syncytial virus (RSV), a member of the family *Pneumoviridae* and genus *Orthopneumovirus* [1], is a common cause of acute lower respiratory infection (ALRI) and a major cause of hospital admissions in young children worldwide. Associated with significant morbidity and mortality, RSV infections place a substantial burden on healthcare services in both high- and low-income countries [2].

RSV can be divided into two subgroups (A and B), although it has only one serotype [3]. To date, 15 RSV-A genotypes and 37 RSV-B genotypes have been defined [4,5]. In regions with a temperate climate, RSV circulation begins in autumn, peaks in winter, and ends in spring [6,7]. Importantly, the predominant RSV subtype and proportion of infections caused by each subtype vary, as does the epidemic size and the timing of the RSV season. According to surveillance data in China, the RSV season onset and peak occur 3–5 weeks earlier and the RSV season duration is 6 weeks longer in RSV-A-prevailing years than those observed in RSV-B-prevailing years. Interestingly, not only can RSV-A and RSV-B predominate alternately, but they can also co-circulate during the same season [8,9]. Therefore, the epidemic onset and offset of an RSV season are difficult to predict because the variability in these timings may depend on temperature, humidity, precipitation, and other environmental and social drivers of seasonality [10].

The clinical manifestations of RSV infection are not sufficiently representative to distinguish this infection from other respiratory viral infections. When only the upper respiratory tract is infected, the main symptoms are mild and include fever, rhinorrhea, cough, and wheezing. However, if the virus invades the lower respiratory tract at the same time, it can cause severe life-threatening lower respiratory tract involvement, such as bronchiolitis and pneumonia [11]. Acute respiratory distress syndrome causes 2–3% of pediatric patients with RSV infection to be admitted to hospitals, and approximately 45% of hospital admissions and in-hospital deaths due to RSV-ALRI occur in children aged younger than 6 months [12]. Severe RSV infection in infants and young children is associated with sustained wheezing, asthma, and lung function decline at school age [13]. The clinical effects of viral factors during RSV infection are not clear, and links between different subtypes and genotypes and disease severity have not been fully confirmed [14,15]. Coinfection with respiratory viral and bacterial pathogens is common [16,17], but there is no consensus on how the clinical severity of the disease caused by RSV infection is influenced by the additional infection with another pathogen [18,19,20].

The prevalence of RSV varies from region to region, and although some studies on the epidemiology of respiratory infections have recently been reported for local areas of China [9,21], there have been few articles describing continuous surveillance across multiple hospitals aimed at understanding the epidemiological and clinical characteristics of RSV infection. Therefore, we analyzed the epidemiological and clinical examination data from hospitals in China collected during the period from 2014 to 2018 to gain insight into the disease severity of RSV infection in pediatric patients (aged <5 years). Such data are critical for the successful implementation of prevention, control, and treatment strategies for RSV, and they will also be helpful in guiding the development of an RSV vaccine.

## 2. Materials and Methods

### 2.1. Patients and Specimens

In total, 19,898 samples from children aged <5 years with an acute respiratory tract infection (ARTI) were collected during active surveillance of hospitalized patients with ARTI in China from January 2014 to December 2018. The inclusion criteria were [17,22]: (1) at least one of the following abnormal detection results: fever, white blood cell (WBC) differentials, leukocytosis or leukopenia; and (2) at least one of the following signs or symptoms: cough, chills, expectoration, nasal congestion, sore throat, chest pain, tachypnea, and abnormal lung rale. The respiratory samples, including nasopharyngeal swabs and aspirates, sputum, bronchoalveolar lavage fluid (BALF), pleural effusion, and blood, were collected prior to the initiation of any therapeutic measures, and they were transported to the infectious disease surveillance laboratories for testing within 24 h of collection or were stored at −80 °C until use.

### 2.2. Detection of Respiratory Pathogens

Eight respiratory viruses, specifically RSV, influenza virus (Flu), parainfluenza virus (PIV), human metapneumovirus (HMPV), human coronavirus (HCoV), human adenovirus (HADV), human rhinovirus (HRV), and human bocavirus (HBoV), were detected by PCR or real-time PCR assays as described previously [21,22]. Briefly, virus DNA and RNA were extracted from 200 µL of respiratory samples using automated nucleic acid extraction equipment (i.e., from Roche (Basel, Switzerland)/Qiagen (Hilden, Germany)/bioMerieux (Marcy-l’Étoile, France)/Applied BioSystems (Waltham, MA, USA)) in accordance with the manufacturer’s instructions. Reverse transcription of viral RNA was performed using commercially available kits (i.e., from Invitrogen/Roche/Qiagen/Promega/Takara), and the real-time PCR procedure was: 95 °C for 10 min, followed by 45 cycles of (95 °C for 15 s, 55–60 °C for 1 min) on an ABI7500 Real-time PCR system.

Blood, nasopharyngeal aspirate, sputum, BALF, and pleural effusion samples were collected for use in the detection of bacteria (*Streptococcus pneumoniae*, *Staphylococcus aureus*, *Klebsiella pneumoniae*, *Pseudomonas aeruginosa*, *Streptococcus pyogenes*, *Haemophilus influenzae*, *Legionella pneumophila*, *Mycoplasma pneumoniae*, and *Chlamydia pneumoniae*) by culture or PCR. In brief, pour freshly prepared nutrient agar medium into the sterile Petri plates and allow it to solidify, take an inoculating loop and sterilize it on the flame until it gets red hot. Then, take the inoculum by using a sterilized inoculating loop and streak over the solid nutrient media by keeping the plate close to the flame to avoid contamination. After streaking, incubate the culture plates for 24–48 h at a temperature of 35–37 °C [23].Urine was also collected for use in rapid antigen assays (Gold Immunochromatography Assay) for detecting *L*. *pneumophila* and *S*. *pneumoniae* according to the detection step kit (Binax NOW^®^ Strep A Test) [24].

### 2.3. Data Collection and Statistical Analysis

Detailed patient demography, epidemiology, clinical information, and laboratory results, including case history, symptoms, physical signs, and examination results, for each patient were collected by the staff of sentinel hospitals and laboratories and then reported through an online data management system established by Chinese Center for Disease Control and Prevention (China CDC, Beijing, China).

Descriptive statistics included frequency analysis for categorical variables such as gender, age group, case type, and RSV subtype. Statistical analysis was performed with Excel software (Microsoft Co., Washington, DC, USA) and SPSS (v18.0, SPSS, Chicago, IL, USA). The normality of the distribution of continuous variables was tested by a one-sample Bayesian test. Continuous variables with normal distribution were presented as mean (standard deviation [SD]) for laboratory tests; age distribution was reported as median (interquartile range [IQR]). Independent samples Student’s *t*-test or One-Way ANOVA test were used, respectively, to compare means of 2 and 3 or more groups of variables normally distributed. Categorical variables were compared with a chi-squared test or Fisher’s exact test. A two-sided *p*-value of <0.05 was considered to indicate statistical significance.

## 3. Results

### 3.1. Clinical and Epidemiological Characteristics of RSV-Positive Cases

From the active surveillance of hospitalized patients with ARTIs in China conducted from 2014 to 2018, we included 19,898 pediatric patients (aged < 5 years). Among the 17.3% (3449/19898) who were found to be positive for RSV, of these 63.9% (2205/3449) were boys and 36.1% (1244/3449) were girls. The median age of the enrolled patients was 0.8 (0.3, 1.8) years. Among the study participants, 35.8% (1234/3449) were aged <6 months, 21.7% were aged 6–12 months, and 20.2% were aged 1–2 years. The RSV incidence in infants (aged < 6 months) was much higher than that in older children (*p* < 0.01). The RSV-positive patients were 89.1% (3073/3349) and 10.9% (376/3349) detected from inpatients and outpatients, respectively (Table 1).

The annual distributions of RSV in China are shown in Figure 1. The annual RSV incidences were 5.4% (1056/19467), 6.4% (777/12175), 7.8% (536/6906), 6.0% (440/7304), and 7.0% (640/9147) from 2014 to 2018, respectively. High RSV incidence was found in the year 2015 (*p* < 0.05), particularly in the month of December. The percentage of positive RSV case had significant difference for age (*p* = 0.025), case type (*p* < 0.001) and RSV subtype (*p* < 0.001), except for sex (*p* = 0.165) in different study years (Table 1). Importantly, infection with RSV can be detected nearly year-round, but it occurs mainly from late autumn through early spring (November to April), with a peak from December to February each year. RSV-positive cases were rarely observed during the summer months (June to August) (Figure 1).

The clinical diagnoses of the study participants are listed in Table 2. The common symptoms of RSV infection were cough (93.2%, 3216/3449), fever (47.7%, 1645/3449), expectoration (40.2%, 1387/3449), lung rale (37.5%, 1285/3449), runny nose (24.9%, 859/3449), breathing difficulty (15.8%, 546/3449), diarrhea (11.7%, 403/3449) and sore throat (6.5%, 225/3449). In addition, compared with other age groups, children aged 1–2 years had a significantly higher incidence of fever (75.6%), and those aged 3–5 years had higher incidences of sore throat (13.4%) and runny nose (31.8%), and children aged <1 year were more likely to develop symptoms of expectoration, breathing difficulty, diarrhea, and lung rale. Laboratory tests revealed that children aged 0–6 months had increased risks of presenting with a reduced white blood cell count (3.8 ± 0.5 × 10^9^/L) and an elevated platelet count (443.9 ± 131.3 × 10^9^/L) compared with the normal range for infants. Meanwhile, compared with older RSV-infected children, RSV-infected infants presented with a higher mean lymphocyte percentage (47.4% ± 19.8%) and a lower mean neutrophil percentage (38.2% ± 19.1%) (*p* < 0.001). There was no significant difference in the oxygen saturation (SaO_2_) among the age-based groups across the 5 years of the study. In hospitalized children, RSV was detected significantly more frequently (46.0%) in children aged 0–2 years with a diagnosis of bronchiolitis or bronchopneumonia as compared with a diagnosis of pneumonia (Table 2).

### 3.2. Epidemiological Distribution of RSV Subtypes

Among typed RSV strains, RSV-A was observed as the prevalent (53.3% (1734/3449)) circulating subtype in China from 2014 to 2018; 29.4% (1014/3449) of RSV strains were found to belong to the RSV-B subgroup, while 20.3% (701/3449) could not be typed (Table 3). Most of the time, the seasonal characteristic observed for RSV-A was similar to that observed for total RSV, with epidemic peaks from December to January and lower peaks from July to September. RSV-B showed a 1-month stagger in its epidemic peaks (Figure 1B). During the study period, RSV-A and RSV-B had dominance alternately: RSV-A was the main subtype in 2014–2015 and 2017–2018, RSV-B was predominant in 2016–2017, and these two subtypes were both epidemic at a high level in 2015–2016. The RSV dominance pattern followed a trend of 2 years of RSV-A dominance and 1 year of RSV-B dominance.

In our RSV-positive cases, a high RSV-A positivity rate (52.2% (644/1234)) was observed in infants aged <6 months; additionally, 52.7% (394/747) of RSV cases in children aged 6–12 months, 49.8% (347/697) of RSV cases in children aged 1–2 years, and 45.3% (349/771) of RSV cases in children aged 3–5 years were RSV-A positive (Table 2). Male patients accounted for 62.8% (1089/1734) of RSV-A cases and 65.0% (659/1014) of RSV-B cases; there was no significant difference in the patient sex between these two RSV subtypes (*p* = 0.250). The incidence of the most common RSV symptoms, i.e., sore throat, expectoration, breathing difficulty, and lung rale, did not differ by RSV subgroup (Table 3); only cough (92.2%), running nose (19.8%), and diarrhea (12.2%) had a lower incidence in the RSV-A subgroup compared with the RSV-B subgroup (*p* < 0.05). The laboratory test results such as WBC (*p* < 0.001) and PLT (*p* = 0.012) had been shown to be significant. In addition, we found no significant differences in the incidence of severe pneumonia between the RSV-A and RSV-B groups, except that RSV-A may be more likely to cause a fever, especially a high fever (≥39.1 °C).

### 3.3. Clinical Manifestations of RSV-Positive Cases in Which Other Common Respiratory Pathogens Were Co-Detected

We assessed the RSV-positive samples for the presence of seven other common respiratory disease-causing viruses (Flu, PIV, HMPV, HCoV, HADV, HRV, and HBoV) and bacteria (*S*. *pneumoniae*, *S*. *aureus*, *K*. *pneumoniae*, *P*. *aeruginosa*, *Streptococcus pyogenes*, *H*. *influenzae*, *L*. *pneumophila*, *M*. *pneumoniae*, and *C*. *pneumoniae*) (Table 4).

#### 3.3.1. Mono RSVs Infection

Among the 3449 patients with RSV, 64.6% (2229/3449) were infected with only RSV. Of these, 8.3% (184/2229) exhibited high fever (≥39.1 °C). The proportion of patients with signs and symptoms such as cough, sore throat, expectoration, running nose, breathing difficulty, diarrhea, and lung rale was 93.8%, 7.1%, 62.2%, 26.9%, 12.3%, 90.1%, and 76.7% respectively. Additionally, abnormal laboratory tests e.g., SaO_2_ (%) was 92.3% ± 11.3, Lymphocyte (%) was 51.9% ± 22.3, Neutrophil (%) was 38.0% ± 17.6, HGB was 114.7 ± 14.9 g/L. The percentage of positive cases hospitalized was 86.0% (1917/2229) and 8.6% (191/2229) of patients finally developed into severe pneumonia.

#### 3.3.2. Co-Infection of RSVs with Other Viruses

RSV is most prevalent in winter, overlapping with the endemic period of other respiratory viruses such as influenza and PIV. In our study, we found that 14.4% (497/3449) patients had detected other viruses additionally, most commonly PIV, followed by Flu and HADV. The proportion of high fever (≥39.1 °C) has increased in patients of co-infection of RSVs with other viruses was 11.9%. The signs and symptoms were 86.7% cough, 8.9% sore throat, 69.0% expectoration, 18.3% running nose, 18.9% breathing difficulty, 91.1% diarrhea, and 77.2% lung rale. That is, 90.9% (452/479) of coinfected patients were detected from inpatients, and the proportion of severe pneumonia was 6.6% (33/479).

#### 3.3.3. Co-Infection of RSVs with Bacteria

There was 5.0% (172/3449) of the patients with RSV had additional detected bacteria, most commonly *S*. *pneumoniae* and *H. influenzae*. In addition, we found a few cases patients of 0.7% (24/3449) and 0.2% (7/3449) were co-infected with mycoplasma and chlamydia. The proportion of high fever (≥39.1 °C) has decreased in patients with co-infection of RSVs with bacteria was 2.9%. The proportion of patients who had a cough, sore throat, expectoration, running nose, breathing difficulty, diarrhea, and lung rale was 97.7%, 0%, 30.8%, 22.1%, 25.6%, 76.7%, and 89.2%, respectively. Additionally, Lymphocyte (%) and Neutrophil (%) was 51.9% ± 22.3 and 38.0% ± 17.6. Importantly, the coinfected patients were 98.8% (170/172) from inpatients and 22.7% (39/172) were severe pneumonia.

#### 3.3.4. Co-Infection of RSVs with Other Viruses and Bacteria

We then detected varieties of viral and bacterial co-infections. That is, 16.0% (551/3449) of RSV-positive patients had both other viruses and bacteria co-detected. The proportion of high fever (≥39.1 °C) was 5.3%. The signs and symptoms were 95.5% cough, 4.0% sore throat, 50.6% expectoration, 23.8% running nose, 24.3% breathing difficulty, 82.2% diarrhea and 77.3% lung rale. The coinfected patients had a high proportion of severe pneumonia (96.9%) and inpatients (14.7%).

We analyzed the clinical manifestations of RSV-positive cases with or without coinfection and found no correlation was found between pathogen co-detection and patient sex (*p* = 0.473). Coinfection with other viruses may be more likely to cause fever compared with RSV infection alone or RSV coinfection with bacteria, given that 11.9% of patients coinfected with other respiratory viruses exhibited high fever (≥39.1 °C). Infants (aged <6 months) were the most likely population to be infected with only RSV (37.3%) or to be coinfected with bacteria (45.3%), whereas children aged 3–5 years were more susceptible to coinfection with other respiratory viruses (29.2%). The pathogen co-detection rates were 38.1% in inpatients (1156/3073) and 15.9% (59/371) in outpatients, with a significant difference between these rates inpatients and emergency/outpatients (*p* < 0.05). Importantly, the occurrence of the highest rate of RSV co-detection with bacteria (98.9%) in inpatients may be related to the high incidence of severe pneumonia in this group, which was significantly higher than that in RSV-single-positive patients (*p* < 0.05).

## 4. Discussion

RSV is one of the major pathogens that cause ARTIs with high morbidity and mortality, especially in children under 5 years of age [25,26,27]. A better understanding of the epidemiological and clinical characteristics of RSV infection is very important for ARTI prevention and control. In this study, by using surveillance data on ARTIs from the mainland of China from 2014 to 2018, we have explored the demographics of patients with RSV (aged < 5 years), identified the epidemiological and clinical characteristic differences in RSV subgroups and assessed RSV coinfection with other respiratory viral and bacterial pathogens.

Like similar studies carried out by other groups [26,28], our study analyzed the sex and age distribution of children (aged < 5 years) who were infected with RSV and found that boys had a high proportion of RSV infection. Additionally, 35.8% of the patients with RSV were aged <6 months, demonstrating that boys and infants were particularly susceptible to RSV infection, although the morbidity needs further analysis. Cough and fever were common symptoms exhibited by the patients with RSV in our study, followed by expectoration, runny nose, and breathing difficulty, which is consistent with the findings of other similar studies [9,29]. In addition, our results show that RSV infections were more commonly detected in inpatients, with more than 3/4 of the 3449 pediatric patients in this study being inpatients. RSV infection was more likely to cause pneumonia and bronchitis [16,22], and by comparing the incidence by age between patients with pneumonia and those without pneumonia, we found that RSV was the most frequent cause of bronchial pneumonia in children aged 0–2 years.

Here, we found that RSV circulated year-round, but its peak incidence occurred during the winter and early spring months, which is consistent with reports from other regions [30,31]. The annual distribution of RSV during 2014–2018 in China uncovered 2015 as a high epidemic peak year, in which both RSV-A and RSV-B were epidemic at high levels. Our RSV subtyping analysis revealed that RSV-A was the main subtype in 2014–2015 and 2017–2018, whereas RSV-B was predominant in 2016–2017. The finding that the RSV year distribution followed a dominance pattern trend of 2 years for RSV-A and 1 year for RSV-B is similar to those reported by others [21,32,33]; patterns of RSV-A dominance for 2–4 years followed by RSV-B dominance for 1 year were also reported in Uruguay and Japan [34,35]. Data generated from previous studies reveal that a complex interaction of climate factors, especially temperature and humidity, influences the epidemic pattern. Notably, this study found no significant differences in the main systemic symptoms or laboratory test results between the RSV-A and RSV-B groups. Although both RSV-A and RSV-B infected mainly young children, a higher RSV-A positivity rate was found in infants (aged 0–12 months), indicating that RSV-A may have a higher pathogenicity for infants.

Coinfection with respiratory viruses and bacteria is a common phenomenon [17,22]. This is because respiratory pathogen activity exhibits a peak, which occurs in the winter/spring seasons [36]. In the present study, 35.4% of the RSV-positive cases RSV had other detectable viruses or bacteria; PIV and *S. pneumoniae* were the most frequently detected coinfection pathogens. Although there are reports that coinfection of respiratory viruses does not aggravate disease severity [21,37], we found that the pathogen co-detection rate, particularly the bacteria co-detected rate, was higher in RSV-infected inpatients. Compared with the RSV alone group, the group coinfected with other viruses had a significantly higher incidence of fever, and the group coinfected with bacteria had significantly higher incidences of breathing difficulty and severe pneumonia, indicating that coinfection might be related to the disease severity for RSV.

This study was subject to several limitations. First, the hospitals in our study are not evenly distributed throughout the mainland of China; there were fewer samples collected in west regional hospitals, which affects the representation of regional epidemiological characteristics. Second, several factors, such as failure to consecutively collect samples and usage of antibiotics or antiviral drugs prior to treatment, might result in low RSV detection rates. Third, viral load detection and RSV genotyping were not performed to explore the molecular epidemiological characteristics of RSV and its relationship with clinical features.

## 5. Conclusions

This study investigated important epidemiological and clinical characteristics of RSV in patients aged <5 years with ARTI in China from 2014 to 2018. Our findings, together with ongoing global studies, may aid in guiding health policies for the prevention and control of RSV infection and may provide important clues for the administration of antivirals or vaccines when available.

## Figures and Tables

**Figure 1 vaccines-10-00810-f001:**
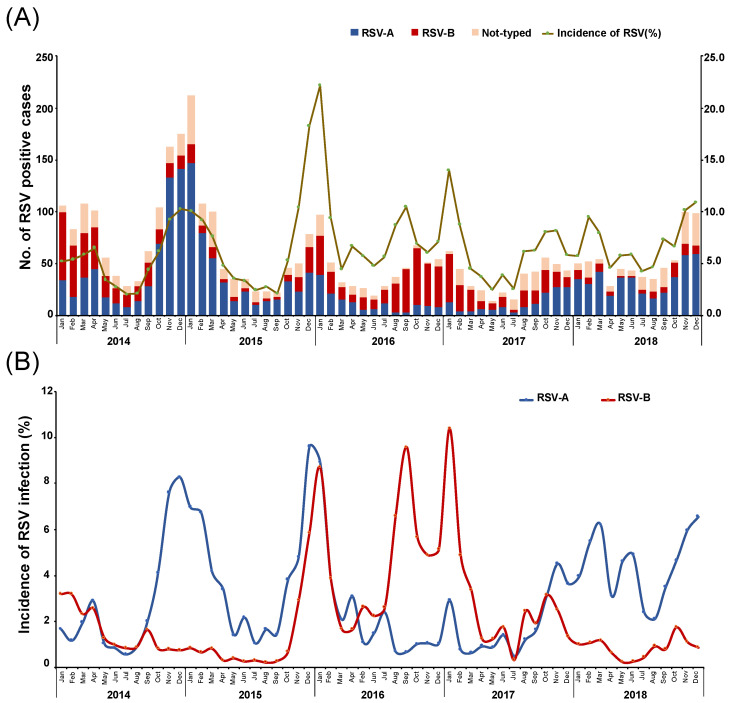
Monthly distribution of RSV-A and RSV-B from January 2014 to December 2018 in China. The number of RSV-A- or RSV-B-positive patients and the monthly RSV incidence (% of RSV-positive patients in the indicated month) are shown. (**A**) The numbers of RSV-A-positive and RSV-B-positive cases; (**B**) the monthly incidences of RSV-A and RSV-B.

**Table 1 vaccines-10-00810-t001:** Demographic characteristics of study participants by year.

Demographic Characteristics	2014 N = 1056	2015 N = 777	2016 N = 536	2017 N = 440	2018 N = 640	*p*
Sex (%)						0.165
Male	679 (64.3)	496 (63.8)	363 (67.7)	279 (63.4)	388 (60.6)	
Female	377 (35.7)	281 (36.2)	173 (32.3)	161 (36.6)	252 (39.4)	
Age, median (IQR, years)	0.8 (0.3,2.0)	0.8 (0.3,1.7)	0.7 (0.3,1.5)	0.7 (0.3,1.5)	0.9 (0.4,1.9)	0.004
Age (%)						0.025
<6 months	379 (35.9)	276 (35.5)	212 (39.6)	169 (38.4)	198 (30.9)	
6–12 months	205 (19.4)	179 (23.0)	110 (20.5)	107 (24.3)	146 (22.8)	
1–2 years	211 (20.0)	159 (20.5)	108 (20.1)	73 (16.6)	146 (22.8)	
3–5 years	261 (24.7)	163 (21.0)	106 (19.8)	91 (20.7)	150 (23.5)	
Case type (%)						<0.001
Outpatients	121 (11.5)	69 (8.9)	44 (8.2)	37 (8.4)	105 (16.4)	
Inpatients	935 (88.5)	708 (91.1)	492 (91.8)	403 (91.6)	535 (83.6)	
RSV subtype						<0.001
RSV-A	555 (52.6)	486 (62.5)	144 (26.9)	138 (31.4)	411 (64.2)	
RSV-B	322 (30.5)	100 (12.9)	317 (59.1)	195 (44.3)	80 (12.5)	
Unclassified type	179 (17.0)	191 (24.6)	75 (14.0)	107 (24.3)	149 (23.3)	

**Table 2 vaccines-10-00810-t002:** Clinical characteristics of RSV-positive children in age group.

Variables N (%)	<6 Months N = 1234	6–12 Months N = 747	1–2 Years N = 697	3–5 Years N = 771	*p*
Sex					<0.001
Male	812 (65.8)	521 (69.7)	445 (63.8)	427 (55.4)	
Female	422 (34.2)	226 (30.3)	252 (36.2)	344 (44.6)	
Signs and symptoms					
Fever	402 (32.6)	397 (53.1)	527 (75.6)	319 (41.4)	<0.001
Cough	1165 (94.4)	699 (93.6)	647 (92.8)	705 (91.4)	0.073
Sore throat	50 (4.1)	35 (4.7)	33 (4.7)	107 (13.9)	<0.001
Expectoration	509 (41.2)	331 (44.3)	271 (38.9)	276 (35.8)	0.006
Running nose	222 (18.0)	193 (25.8)	199 (28.6)	245 (31.8)	<0.001
Breathing difficulty	276 (22.4)	129 (17.3)	75 (10.8)	66 (8.6)	<0.001
Diarrhea	190 (15.4)	134 (17.9)	61 (8.8)	18 (2.3)	<0.001
Lung rale	480 (81.9)	286 (78.6)	261 (74.8)	258 (71.7)	0.009
Laboratory tests					
SaO_2_ (%)	93.9 ± 8.7	92.9 ± 9.5	93.6 ± 6.1	93.3 ± 11.4	0.681
WBC (×10^9^/L)	3.8 ± 0.5	4.4 ± 0.1	4.6 ± 0.1	4.6 ± 0.1	<0.001
Lymphocyte (%)	47.4 ± 19.8	44.2 ± 22.0	49.2 ± 29.7	33.1 ± 18.6	<0.001
Neutrophil (%)	38.2 ± 19.1	37.1 ± 22.0	45.5 ± 18.2	48.8 ± 16.4	<0.001
HGB (g/L)	107.6 ± 13.1	112.6 ± 16.1	121.0 ± 15.1	122.1 ± 12.5	<0.001
PLT (10^9^/L)	443.9 ± 131.3	368.3 ± 97.6	288.0 ± 175.9	305.9 ± 102.0	<0.001
RSV subtype					<0.001
RSV-A	644 (52.2)	394 (52.7)	347 (49.8)	349 (45.3)	
RSV-B	407 (33.0)	227 (30.4)	194 (27.8)	186 (24.1)	
Unclassified type	183 (14.8)	126 (16.9)	156 (22.4)	236 (30.6)	
Clinical diagnosis					<0.001
Pneumonia	469 (38.0)	263 (35.2)	254 (36.4)	282 (36.6)	
Bronchopneumonia	579 (46.9)	334 (44.7)	319 (45.8)	281 (36.4)	
Others	186 (15.1)	150 (20.1)	124 (17.8)	208 (20.7)	

**Table 3 vaccines-10-00810-t003:** Clinical characteristics of RSV-A positive and RSV-B positive children.

Variables N (%)	RSV-A Positive N = 1734	RSV-B Positive N = 1014	*p*
Sex			0.250
Male	1089 (62.8)	659 (65.0)	
Female	645 (37.2)	355 (35.0)	
Fever (°C)			0.009
<37.3	919 (53.0)	583 (57.5)	
37.3–39.1	666 (38.4)	372 (36.7)	
≥39.1	149 (8.6)	59 (5.8)	
Signs and symptoms			
Cough	1599 (92.2)	956 (94.3)	0.041
Sore throat	97 (5.6)	61 (6.0)	0.647
Expectoration	748 (43.1)	417 (41.1)	0.303
Running nose	343 (19.8)	242 (23.9)	0.012
Breathing difficulty	319 (18.4)	196 (19.3)	0.546
Diarrhea	212 (12.2)	152 (15.0)	0.039
Lung rale	646 (79.5)	320 (78.2)	0.621
Laboratory tests			
SaO_2_ (%)	93.4 ± 8.5	94.2 ± 9.0	0.200
WBC (×10^9^/L)	4.1 ± 0.7	4.5 ± 0.5	<0.001
Lymphocyte (%)	48.5 ± 19.8	42.1 ± 19.5	0.897
Neutrophil (%)	39.8 ± 19.4	44.0 ± 19.0	0.250
HGB (g/L)	110.8 ± 16.7	119.2 ± 12.9	0.197
PLT (10^9^/L)	379.4 ± 166.2	353.2 ± 124.8	0.012
Pneumonia			
Severe	112 (13.8)	53 (13.0)	0.335

**Table 4 vaccines-10-00810-t004:** Clinical characteristics of patients in whom RSV was co-detected with other common respiratory viruses and bacteria.

Variables N (%)	RSV Only N = 2229	With Virus N = 497	With Bacteria N = 172	With Virus–Bacteria N = 551	*p*
Sex					0.473
Male	1407 (63.1)	330 (66.4)	108 (62.8)	360 (65.3)	
Female	822 (36.9)	167 (33.6)	64 (37.2)	191 (34.7)	
Age					<0.001
<6 months	832 (37.3)	128 (25.8)	78 (45.3)	196 (35.6)	
6–12 months	442 (19.8)	119 (23.9)	38 (22.1)	148 (26.9)	
1–2 years	439 (19.7)	105 (21.1)	37 (21.5)	116 (21.1)	
3–5 years	516 (23.1)	145 (29.2)	19 (11.0)	91 (16.5)	
Fever					<0.001
<37.3	1088 (48.8)	205 (41.2)	136 (79.1)	375 (68.1)	
37.3–39.1	957 (42.9)	233 (46.9)	31 (18.0)	147 (26.7)	
≥39.1	184 (8.3)	59 (11.9)	5 (2.9)	29 (5.3)	
Signs and symptoms					
Cough	2091 (93.8)	431 (86.7)	168 (97.7)	526 (95.5)	<0.001
Sore throat	159 (7.1)	44 (8.9)	0 (0)	22 (4.0)	<0.001
Expectoration	1387 (62.2)	343 (69.0)	53 (30.8)	279 (50.6)	<0.001
Running nose	599 (26.9)	91 (18.3)	38 (22.1)	131 (23.8)	0.001
Breathing difficulty	274 (12.3)	94 (18.9)	44 (25.6)	134 (24.3)	<0.001
Diarrhea	2008 (90.1)	453 (91.1)	132 (76.7)	453 (82.2)	<0.001
Lung rale	878 (76.7)	146 (77.2)	74 (89.2)	55 (77.3)	0.075
Laboratory tests ^#^					
SaO_2_ (%)	92.3 ± 11.3	94.2 ± 4.9	95.4 ± 3.7	95.4 ± 4.0	0.998
WBC (×10^9^/L)	4.2 ± 0.7	4.2 ± 0.7	4.0 ± 0.4	4.3 ± 0.6	0.308
Lymphocyte (%)	51.9 ± 22.3	33.3 ± 19.5	53.8 ± 17.0	40.2 ± 19.8	<0.001
Neutrophil (%)	38.0 ± 17.6	54.3 ± 22.1	38.6 ± 17.6	42.3 ± 18.4	0.002
HGB (g/L)	114.7 ± 14.9	111.8 ± 16.0	115.2 ± 13.0	115.8 ± 16.0	0.032
PLT (10^9^/L)	385.2 ± 140.3	364.3 ± 118.9	430.2 ± 146.3	329.2 ± 163.9	<0.001
Case type					<0.001
Outpatients	312 (14.0)	45 (9.1)	2 (1.2)	12 (3.1)	
Inpatients	1917 (86.0)	452 (90.9)	170 (98.8)	534 (96.9)	
Pneumonia					
Severe	191 (8.6)	33 (6.6)	39 (22.7)	81 (14.7)	<0.001

^#^ Laboratory tests: percentage of SaO_2_ (oxygen saturation), and blood routine test including WBC (white blood cell) count, percentage of lymphocyte and Neutrophil, HGB (Hemoglobin) and PLT (platelet count).

## Data Availability

The datasets generated and analyzed during this study are not publicly available owing to the institute’s data security and sharing policy, but they are available from the corresponding author on reasonable request.

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
