# Peer review of "Epidemiological and Clinical Characteristics of Respiratory Syncytial Virus Infections in Children Aged <5 Years in China, from 2014–2018"

_vaccines, 2022, doi:10.3390/vaccines10050810_

Round 1

Reviewer 1 Report

The paper titled “Epidemiological and clinical characteristics of respiratory syncytial virus infections in children aged <5 years in China, from 2014–2018” analyzed the RSV diagnostic testing results from the surveillance of acute respiratory infections in China between 2014 and 2018 to understand the epidemiological and clinical characteristics of RSV in children aged <5 years. This study aims to provide the findings and clues for guiding health policies for the prevention and control of RSV infection and for the administration of antivirals or vaccines when available.

Concerns:

The statistical analyses are not sound and the explanations for the data are confusing and misleading. I suggest that the authors find a statistician to help with the statistical analyses and explanations for the data.

  1. Line 19, “RSV infection was significantly more common among individuals aged <6 months (n = 1234; 35.8%).” This explanation should be based on the statistical analyses, but where are the analyses and the p value? The same issue exists in line 22, “Cough was the most universal (93.1%) symptom of RSV infection”. The authors should indicate how the value of 93.1% was calculated and also provide the p value. Line 24, “There were no significant differences in systemic symptoms between the RSV-A and RSV-B groups.”. The authors need to clarify what the systemic symptoms are. Cough, running nose and diarrhea are significant between the RSV-A and RSV-B groups.

  1. T-test is a parametric test but Kruskal-Wallis test is a nonparametric test. Why didn’t the authors apply the same type of test in the analyses?

  1. Line 123, “Among the 3449 (17.3%) who were found to be positive for RSV, 2205 (63.9%) were boys and 1244 (36.1%) were girls; this difference was statistically significant (p < 0.01).” The authors should indicate how they computed the value of 17.3% as this data is not shown in the tables. Additionally, what method did they use to compare these two proportions, 63.9% and 36.1%? Chi-square is used to analyze the association between two variables. These are the proportions of one variable. Line 126, “35.3% (1234/16024) were aged <6 months,” is not correct and it should be 35.8% (1234/3449), the same as table 1. P values should be presented in table 1 as the text uses the p values to explain the data. There is the same issue with the comparison of the 89.1% vs. 10.9% values, p < 0.01. The explanation of the results should be based on the data in the table. In table 1, add (%) after sex, age and case type.

Line 142, “The most common symptoms of RSV infection were cough (93.2%), fever (47.7%), expectoration (40.2%), runny nose (24.9%), breathing difficulty (15.8%), diarrhea (11.7%), and sore throat (6.5%);”. Again, please demonstrate how these data were calculated. Line 144 “additionally, 77.5% of patients showed signs of lung rale, which was not a distinguishing symptom from those of infections with other common respiratory viruses.”. It seems as if the authors were discussing the data from other tables rather than table 2. Line 154, “There was no significant difference in the oxygen saturation (SaO2) among the age-based and sex-based groups across the 5 years of the study.”. The sex-based SaO2 data are not found in the paper.

Line 163, “Among typed RSV strains, RSV-A was observed as the prevalent (53.3% [1734/3449]) circulating subtype in China from 2014–2018; 29.4% (1014/3449) of RSV strains were found to belong to the RSV-B subgroup, while 20.3% (701/3449) could not be typed (p < 0.001) (Table 3).” These data haven’t been shown in Table 3 and how the authors computed the p value is also not demonstrated. Line 186, “only cough (97.4%),” it is 92.2% in Table 3. Line 187 “We found no significant differences in the laboratory test results”. WBC and PLT have been shown to be significant in table 3. Line 199, please show the calculation of the 64.6% and 14.4% values. Line 213, the value of 29.9% is different from table 3. Line 214, the 1156/3037 value is different from table 3. There are also English typos in the tables, such as “Breach difficulty”.

  1. The authors need to revise the discussion as well based on the data shown and explain the data carefully otherwise it is confusing and misleading. The authors should make sure that they understand the row percentage and column percentage in the Chi-square analysis. In addition, the authors also need to indicate clearly how the data are calculated in the text, especially for the data that are not shown in the tables.

In brief, there are numerous errors in the paper and the authors should check and revise the paper carefully and thoroughly and include sound statistical analyses.

Reviewer 2 Report

Zhang et al. analyzed the epidemiological characteristics of Chinese children under the age of 5 hospitalized with RSV inspection from 2014 to 2018. It will be a crucial observational study in public health management and policy-making for RSV, the most common cause of acute lower resistance in children in East Asia.

Globally, mycoplasma is the second most common cause of pneumonia in children after RSV, so it would be more interesting to suggest the frequency of co-infected bacterias.

In addition, this study has reported that RSV is most prevalent in winter, which can overlap with the endemic period of influenza, so the presence of RSV-influenza accompanying infection will also be significant in terms of public health as vaccines.

As a minor concern, ARTI and ARI are abbreviations for acute lower resolution information, so please express them as one of the two.

Additional descriptions of epidemiological issues that may be interesting and important in clinical and public health policies have been recommended.

Reviewer 3 Report

The authors presented an interesting and necessary article that complements the numerous works that describe the features of the spread of the RS–virus in various regions of the world in the pre–pandemic period for COVID–19. This allows us to create a more complete picture of its distribution in the world. Also, the presented results may shed light on a still unclear phenomenon – why later SARS–CoV–2 replaced most respiratory viruses, but could not dramatically affect the circulation of viruses such as RS–viruses and rhinoviruses.

Point 1. Line 33–34. Talking about the taxonomy of the RS virus, it is better to refer to the official website of the International Committee on Taxonomy of Viruses (ICTV) <https://talk.ictvonline.org/taxonomy/> but not to the review article [1].

Point 2. Line 88. Please, replace “…specifically RSV and influenza virus (Flu)…” with “…specifically RSV, influenza virus (Flu)…”

Point 3. Line 103. Please, add brief information how did you culture samples for bacteria isolation and how urine was tested.

Point 4. It is well known, that mixed etiology of pulmonary infections is frequent, with co–infection of pathogenic respiratory viruses and bacteria in respiratory secretions. Typically, mixed infections appear to show a more severe clinical condition then monoinfections. For clarity, I suggest to split section 3.3 of the Results to three additional subsections, namely “mono RS–virus infection,” “Co–infection of RS–virus with other viruses” and “Co–infection of RS–virus with bacteria” and present appropriate results separately, subsection by subsection.

Point 5. Line 195–198. The listing of pathogens is a repeat of section 2.2 (Lines 88–90 and 100–104) and may be deleted.

Point 6. Line 193–194. Why the authors never used such common terms as mono– and mixed infections? The title of this subsection (“Clinical manifestations of RSV‐positive cases in which other common respiratory pathogens were co–detected”) is confusing. The alternative title, like, for instance, “Clinical manifestations of mono– and mixed infections,” would be much clear.

Point 7. Line 219 (Title of Table 4). Please, replace “co–detected” with “co–infected”

Point 8. Table 4. The column headings should be changed. Please, use “RS–virus monoinfection,” “RS–virus mixed infection with...”

Point 9. Line 278–283 (Conclusions). Please, after the first sentence, add one more containing your most important results. In that case, your next sentence ("Our findings, together with...") would seem logical.

Round 2

Reviewer 1 Report

I believe that the authors applied ANOVA or Kruskal-Wallis to analyze the data that have >2 groups. Please indicate this in the methods.

Author Response

Point : I believe that the authors applied ANOVA or Kruskal-Wallis to analyze the data that have >2 groups. Please indicate this in the methods.

Response : Thanks for your great suggestion on improving the accessibility of our manuscript. We had corrected the description of methods. “Measurement data are presented as the mean ± SD and were analyzed using an unpaired Student’s t-test (two groups) or one-way ANOVA test (> two groups).”

We would like to take this opportunity to thank you for all your time involved and this great opportunity for us to improve the manuscript. We hope you will find this revised version satisfactory.

Sincerely,

The Authors